# Treatable Traits in Chronic Respiratory Disease: A Comprehensive Review

**DOI:** 10.3390/cells10113263

**Published:** 2021-11-22

**Authors:** Yong Qin Lee, Asvin Selvakumar, Kay Choong See

**Affiliations:** 1Yong Loo Lin School of Medicine, National University of Singapore, Singapore 119077, Singapore; leeyongqin@u.nus.edu (Y.Q.L.); kay_choong_see@nuhs.edu.sg (K.C.S.); 2Division of Respiratory & Critical Care Medicine, Department of Medicine, National University Hospital, Singapore 119228, Singapore

**Keywords:** treatable trait, chronic respiratory disease, multidisciplinary, personalised medicine

## Abstract

Chronic respiratory diseases are major contributors to the global burden of disease. While understanding of these diseases has improved, treatment guidelines have continued to rely on severity and exacerbation-based approaches. A new personalised approach, termed the “treatable traits” approach, has been suggested to address the limitations of the existing treatment strategies. We aim to systematically review the current evidence regarding treatable traits in chronic respiratory diseases and to identify gaps in the current literature. We searched the PubMed and Embase databases and included studies on treatable traits and chronic respiratory diseases. We then extracted information on prevalence, prognostic implications, treatment options and benefits from these studies. A total of 58 papers was included for review. The traits identified were grouped into five broad themes: physiological, biochemical, psychosocial, microbiological, and comorbidity traits. Studies have shown advantages of the treatable traits paradigm in the clinical setting. However, few randomised controlled trials have been conducted. Findings from our review suggest that multidisciplinary management with therapies targeted at treatable traits has the potential to be efficacious when added to the best practices currently implemented. This paradigm has the potential to improve the holistic care of chronic respiratory diseases.

## 1. Introduction

Chronic respiratory diseases are pathologies of the airways and respiratory tract, for example, asthma, chronic obstructive pulmonary disease (COPD), and bronchiectasis. While new developments in our understanding of these diseases have been documented widely in literature, general principles of treatment proposed by various guidelines have remained consistent over the years, relying on severity and exacerbation-based approaches. While these approaches are largely effective in chronic-disease control, it is recognised that patients with heterogenous diseases consisting of multiple phenotypes, like asthma, do not respond equally to the same type of treatment and have differing risks of exacerbation and prognosis [1]. Hence, a relatively recent treatment strategy proposed to tackle these limitations is a “treatable traits” approach.

The “treatable traits” approach involves treating specific traits possessed by patients, defined by specific trait markers, which are subsequently targeted by specific therapies. This differs from the current broad, stepwise treatment of the disease entity, such as the Global Initiative for Asthma guidelines, which has been shown to be limited in its ability to predict exacerbations when used by itself [1]. Recent literature on asthma has shown that multidimensional assessment incorporating treatable traits has the potential to improve care and provide more individualised therapy, allowing management principles to be tailored to each patient’s characteristics or traits [2].

With an increasing number of studies on the treatable traits paradigm reporting its potential benefits, it is important to synthesise the results to help with its clinical applicability. Therefore, we aim to review current literature on the concept of treatable traits, delineate the information available on treatable traits and the prevalence of these traits among various subgroups, and identify treatment and prognostic implications of treating these traits. Through this review, we also sought to identify gaps in the current literature and to provide ideas for further research.

## 2. Materials and Methods

### 2.1. Search Strategy and Selection Criteria

This study has been registered with PROSPERO (registration number: CRD42021250992) and was conducted using the Preferred Reporting Items for Systematic Reviews and Meta-Analysis (PRISMA) guidelines. Two authors (Y.Q.L., A.S.) independently and systematically searched PubMed and Embase for all relevant articles published from inception to 24 October 2020 using the search term “treatable AND trait*”, with the data supplemented by reference-list checks. Studies regarding treatable traits in chronic respiratory diseases were shortlisted, and conference abstracts and review papers were included only if they contained new data not found elsewhere. Non-English papers, papers on acute respiratory diseases such as acute respiratory distress syndrome, and papers with irrelevant topics were excluded from this review. Disagreements between Y.Q.L. and A.S. were discussed with a third author (K.C.S.).

### 2.2. Data Extraction and Quality Assessment

Data on the following were extracted: author, publication date, study type, country, chronic respiratory disease studied, treatable traits, trait-identification marker, prevalence of trait, treatment description, treatment benefits and implications, and prognostic implications. The included studies were independently assessed by 2 authors (Y.Q.L., A.S.) for risk of bias using the Newcastle-Ottawa quality assessment scale (NOS). Disagreements between Y.Q.L. and A.S. were discussed with a third author (K.C.S.). Qualitative synthesis of the studies was conducted.

## 3. Results

A total of 58 studies was included in this systematic review (flow diagram, Figure 1), and quality assessment was performed using the NOS (Table 1). Treatable traits findings from the included studies were grouped into five main themes: physiological traits, biochemical traits, psychosocial traits, microbiological traits, and comorbidity traits.

### 3.1. Physiological Traits

For physiological treatable traits (Table 2), these were studied mostly in the setting of asthma, COPD, and bronchiectasis. These studies described traits including airflow limitation, hypoxemia/hypercapnia, lung hyperinflation, and ciliary dysfunction.

The most prominent physiological trait is airflow limitation, being the most common trait in asthma and COPD. The average prevalence was 52.5% in asthma [5,18,49], with the prevalence varying from 45.5–54.5% between studies [5,18,49]. As defined by a post-bronchodilator forced expiratory volume in the first second < 80% predicted, the prevalence in COPD was the highest, at 88.9% [18]. Airflow limitation has prognostic implications on asthma, leading to worsened severity of symptoms and increased exacerbations and healthcare utilisation [1]. Treatment of airflow limitation in asthma with long-acting muscarinic antagonists (LAMA) and bronchial thermoplasty have been shown to decrease exacerbations, improve lung function, and improve disease control [3,19,39], while short-acting anticholinergics and magnesium were found to decrease hospital admissions [1]. Similar treatments with LAMA, long-acting beta-agonists (LABA), inhaled corticosteroids (ICS), and pulmonary rehabilitation have been proposed to treat airflow limitation in COPD [18,42]. Airflow limitation was also described in bronchiectasis, with inhaled saline and airway clearance being the suggested treatment and with an ongoing trial on epithelial sodium channel inhibition [47].

Lung hyperinflation was also a trait targeted in the management of COPD patients, with endobronchial valves and bronchoscopic thermal vapour ablation showing improved lung function and quality of life [56]. However, prevalence results of lung hyperinflation were not detailed in any of the studies.

### 3.2. Biochemical Traits

Type-2 inflammation is widely documented as a treatable trait in chronic airway diseases like asthma and COPD (Table 3), with a prevalence of 42.0% in asthma patients [49]. Type-2 inflammation is characterised by T2-high expression, such as IL-13 induced genes or high eosinophil count. Chung et al. [59] reported that asthma patients with type-2 inflammation were predisposed to corticosteroid insensitivity and may be dependent on oral corticosteroids in severe cases. Treatment options include bronchodilators containing inhaled corticosteroids (ICS), which have been shown to be effective for asthma. ICS-LABA and ICS-LABA-LAMA combinations have similarly been effective for patients with COPD [42]. Anti-T2 biologics have also been shown to significantly reduce the risk of severe asthma exacerbation and improve lung function and quality of life of patients [59].

Specifically, eosinophilia is one of the most prominent biochemical traits studied, with its prevalence ranging from 51.4% to 56.4% in asthma and 22.2% to 60.1% in COPD [18,29,36]. Eosinophil levels can be measured using blood or sputum eosinophils, and high eosinophil levels are associated with increased risk of exacerbations in both asthma and COPD. In particular for asthma, Feng et al. [10] showed that patients with exacerbation-prone asthma had increased blood eosinophil and reduced lung function at baseline. Treatment for asthma and COPD patients with eosinophilia involves corticosteroid usage, which has been shown to be beneficial in acute exacerbations and to improve FEV1 of patients. Among patients with eosinophilia, monoclonal antibodies, such as anti-IL5, have also been reported to reduce exacerbations in COPD [29]. Specifically for asthma, omalizumab has been shown to significantly reduce both eosinophilia and exacerbations, with improved lung function and quality of life [45].

Neutrophil-related inflammation was also prominent in patients with asthma, bronchiectasis, and COPD, and it is identified by high sputum neutrophil counts [18]. It was associated with increased exacerbation risk for each of the above conditions [30], and high sputum neutrophil counts predicted more rapid decline in FEV1 in patients with bronchiectasis. Macrolides were shown to reduce exacerbations in asthma and COPD, while smoking cessation was additionally recommended for asthma patients. Neutrophil elastase inhibitors were also shown to improve lung function and quality of life for patients with bronchiectasis [4].

As for environmental exposure in united airway diseases identified by immunoglobulin E (IgE) levels and skin-prick tests, proposed treatment options included exposure avoidance, respiratory protection devices, and anti-IgE therapy [57]. However, data on the prevalence of these traits were lacking, and studies evaluating the clinical benefits of the proposed treatment strategies were not available either.

### 3.3. Psychosocial Traits

Two main psychosocial traits identified were treatment adherence/technique and smoking history (Table 4). These two treatable traits were identified across papers studying asthma, COPD, and chronic airway diseases in general. Suboptimal adherence to asthma treatment, as well as poor inhaler technique, was found to be present in 26.9–61.8% of patients [5,18], with a mean of 44.0%. McDonald et al. [30] also found that inhaler-device polypharmacy was more prevalent in severe asthma patients as compared to non-severe asthma patients, and polypharmacy was one of the best predictors of asthma exacerbation risk among 23 other traits. Education and medication reconciliation are the main treatment options, targeting inhaler technique, self-management action plans, and minimising the number of inhaler devices [18].

Smoking was prevalent among patients with chronic respiratory diseases, with a mean prevalence of 14.3% in asthma and 19.4% in COPD [18]. Smoking status was usually assessed through interview or the use of exhaled carbon monoxide and was shown to be a significant risk factor for exacerbation in chronic airway diseases [31]. Smoking cessation was shown to reduce lung-function decline and risk of recurrent exacerbations, highlighting the importance of opportunistic implementation of smoking cessation strategies during acute exacerbations [31]. Management options include counselling with pharmacologic adjuncts, such as nicotine replacement therapy, varenicline, and bupropion [18,31,42].

Low socioeconomic status and poor family support in chronic airway disease were also associated with increased symptomatic deterioration and exacerbations, with social support services proposed as a possible treatment option [31].

### 3.4. Microbiological Traits

Microbiological treatable traits were mostly found in asthma, bronchiectasis, COPD, and united (combined upper and lower) airway diseases (Table 5). The most prevalent trait was chronic respiratory infection in both asthma and COPD, with an average prevalence of 45.0% and a prevalence range of 34.8–47.3% among the studies on asthma [18,21,49]. Various treatment options, such as antibiotics, mucolytics, roflumilast, education, and inhaled interferon-β treatment, were suggested [1,5,18]. However, evidence was lacking regarding their efficacy. In patients with COPD, chronic respiratory infection was present in 55.6% of the patients [18], with prognostic implications on their quality of life and dyspnoea severity [28]. Macrolides have been shown to decrease hospital admissions resulting from exacerbations, but their usage must be balanced against the risk of colonisation with macrolide-resistant organisms [28].

Another prominent microbiological trait is microbial colonisation, present in an average of 18.9% of asthmatics, with a range varying from 12.7 to 55.6% [5,18]. Microbial colonisation was present in an average of 44.8% of COPD patients, with prevalence ranging from 38.9 to 45% [18,27]. This is a separate trait from infection, as colonisation refers to the presence of organisms, while infection refers to the presence of signs and symptoms due to these organisms. Microbial colonisation nonetheless had prognostic significance, demonstrated particularly in COPD patients, with implications on quality of life and increment in dyspnoea [27]. Treatment options for this trait in patients with asthma include education and antibiotic-based written action plans (e.g., using macrolides) [18,19]. However, information regarding treatment options for this trait was inadequate in COPD. Interestingly, patients with microbial colonisation had lower mortality for exacerbations associated with viral infections compared to bacterial infections, though the clinical significance of this is uncertain [27].

### 3.5. Comorbidity Traits

Impaired physical function is a comorbidity seen in almost all chronic respiratory conditions, with a prevalence of 36.1% in COPD and 10.9% in asthma [18]. Impaired physical function can present in different forms, such as low appendicular skeletal muscle mass, limitations in mobility, and low muscle strength and has been shown to be an independent predictor of hospital admission and mortality in COPD. Patients with lower 6 min walking distance were also reported to have higher readmission risk [31]. Various management measures targeted at physical function can be implemented. For instance, Hiles et al. [18] recommended a high-protein diet, strength training, and regular pulmonary rehabilitation for patients with low appendicular skeletal muscle mass.

Another prevalent comorbidity trait is the presence of psychiatric conditions, particularly depression and anxiety in asthma and COPD. Among asthma and COPD patients, anxiety was found in 40% and 27.8%, respectively, while depression was found in 30.9% and 27.8%, respectively [18]. These psychiatric conditions were identified through questionnaires, such as the Hospital Anxiety and Depression scale [18,28]. A study by Matsunaga et al. [28] suggested that psychiatric comorbidities resulted in poorer quality of life, poorer adherence to treatment plans, increased exacerbations, increased hospitalisation, and increased healthcare cost. Treatments included counselling, cognitive behavioural therapy, and paroxetine [18,28].

Nutrition (underweight) is another common comorbidity, with a prevalence of 52.8% in COPD [18] and 38.1% in asthma, with a range of 35.1–58.2% [18]. These traits had similar prognostic implications, with decreased quality of life, increased severity of symptoms, and increased exacerbations [28,30,39,51]. Treatments to normalise body mass index, such as supplementation and weight loss, have also been shown to decrease asthma severity [39]. However, treatment benefits for COPD were not described in the literature.

Some of the other comorbidity traits with high prevalence include systemic inflammation, which was present in 56.4% of asthmatics and 63.9% of COPD patients, and sleep disorders, with a prevalence of 60.0% and 30.6% in patients with asthma and COPD, respectively [18]. For the mitigation of systemic inflammation, McDonald et al. showed that statin therapy improved C-reactive protein levels in COPD patients [2]. Separately, positive airway pressure has been shown to reduce overall mortality and exacerbation risk in COPD patients with sleep disorders, though its efficacy may be limited by patient adherence [28].

## 4. Discussion

In this review of 58 papers, our main findings include the identification of various treatable traits that can be broadly grouped into five themes: physiological, biochemical, psychosocial, microbiological, and comorbidity traits. Specific trait markers were identified for each trait, along with suggested treatment options, benefits and implications of these treatments, and the prognostic implications of the traits. The existence of multiple treatable traits in patients with chronic respiratory diseases suggests that adopting a holistic “treatable traits” approach would benefit clinical care and potentially improve outcomes.

Current literature on the treatable traits paradigm broadly categorises traits into three domains: pulmonary, extrapulmonary, and behavioural [60]. While this approach may be considered broader and more practical, the five-thematic-traits approach proposed in this paper provides a more structured and comprehensive categorisation of treatable traits. Traits of similar pathophysiology are grouped together, along with their therapeutic options, which can help organise treatment plans in the clinical setting for individualised care. The five-theme approach also helps to identify and classify traits that do not fit under the original three domains, allowing these traits to be addressed and considered as part of personalised medicine for patients.

Previous studies have been conducted for specific respiratory conditions, such as asthma and COPD, proving the effectiveness of the treatable-traits algorithm in each of the diseases. This review serves to provide a broad overview of literature regarding treatable traits across chronic respiratory conditions in order to highlight the role and potential of this paradigm on a broader scale. The review also aims to delineate the various gaps in the spectrum of conditions covered to facilitate future research.

### 4.1. Clinical Implications

Certain specific traits can be identified in patients with chronic respiratory diseases, and treating these traits provides benefits. The identification markers of these traits and their treatment were also generally uniform across different diseases, facilitating the formulation of standardised management plans. The treatable traits approach can be organised according to our five identified themes, which can be incorporated into future guidelines for various chronic respiratory diseases.

### 4.2. Clinical Evidence for the Treatable Traits Paradigm

A randomised controlled trial (RCT) was conducted by McDonald et al. [2], comparing management plans targeting pre-defined treatable traits against the current standard of care for severe asthma, where the intervention group reported improvements in health-related quality of life (HRQoL) and reduced exacerbations. Similar findings were found in a study on COPD, where the intervention group achieved significant quality-of-life improvement [61]. While there have been limited RCTs investigating the clinical utility of treatable traits, trials conducted have uniformly reported the benefits of treatable traits in the clinical setting.

As reported in the RCT on severe asthma, medication review led to improved lung function, and breath retraining improved respiratory symptoms [2]. The study also suggested that more emphasis should be placed beyond biomarkers; genetic, phenotypic, and psychosocial traits should also be looked into as part of a holistic management approach. In asthmatic patients, a multidimensional intervention has been shown to have a compounded effect on the HRQoL, demonstrating the potential benefits of a treatable-traits algorithm [2].

Another notable study by Hiles et al. [18] recognised the lack of feasibility in conducting RCTs. This study instead used a Bayesian model-averaging technique on data from two trials, identified traits that were prevalent and had the most significant impact on quality of life, and proposed treatment options to target these traits. Another study that used a similar Bayesian technique was by McDonald et al. [30], studying patients identified from an asthma registry for 24 months to draw comparisons between traits and exacerbation risks.

### 4.3. Gaps in the Current Literature

A treatable trait can be defined as a therapeutic target identified by phenotypes or endotypes through a validated biomarker [31]. However, in our data-collection process, it was noted that some studies used the term ‘treatable traits’ without indicating a clear trait-identification marker or specific therapeutics. Future studies on treatable traits should take into account the definition of treatable traits and include a specific set of trait markers, as well as specific therapeutic measures.

We also note that there was a lack of RCT evidence for treatable traits, and statistics regarding prevalence, treatment efficacy, and prognostic utility were also limited. Future studies, especially RCTs, will be useful to help evaluate the benefits and downsides of the treatable traits approach. Studies on prevalence and prognostic information of treatable traits will also be useful, as traits which are more prevalent or have greater impact on health can be prioritised, for which further studies can be done to explore and improve the management of these specific traits.

Despite its potential in delivering personalised care to patients, further studies have to be conducted to improve the clinical applicability and utility of the treatable-traits algorithm. Different traits with distinct trait-identification markers and therapeutic options can have overlapping pathways in their pathophysiology. Treating each trait with its individual therapeutic options may inadvertently lead to excessive treatments being implemented. One such instance is the relationship between neutrophilic inflammation and smoking, which are currently considered to be distinct traits in existing literature. However, smoking can also be a causative factor for neutrophilic inflammation. Current literature proposes macrolides as a targeted therapy for neutrophilic inflammation in asthma, in addition to smoking cessation. Given the overlap in pathophysiological pathways of the traits, smoking cessation in itself could possibly address both traits, hence avoiding unnecessary treatment [39]. In addition to the thematic approach, which facilitates the use of a common therapeutic agent to target multiple related traits, further research can be conducted to evaluate and integrate traits that can be targeted by common therapeutics.

### 4.4. Future Directions for Research

One future direction for research would be to conduct RCTs, given the lack of RCTs supporting a treatable traits approach. Studying the effectiveness of the treatable traits approach against the current stepwise approach would provide a more conclusive picture of whether there are any advantages conferred over the current approach. As an extension to efficacy trials, real-world effectiveness trials in both primary- and specialist-care settings can be conducted to assess the cost-effectiveness and feasibility of adopting a treatable traits approach. Given that prevalence data from the existing literature are limited, another possible area of research would be to conduct population-based surveys of treatable traits, as these can help direct efforts to tackle the more prevalent traits.

### 4.5. Strengths and Limitations

We have performed a systematic review addressing treatable traits across chronic respiratory diseases, providing a broad overview of current evidence. With inclusion criteria inclusive of all chronic respiratory conditions and a large number of papers reviewed, it provides comprehensive prevalence statistics across conditions and traits and brings new insights into the utility of treatable traits in the clinical setting. In particular, we found that even though a particular treatable trait (e.g., eosinophilia) can be found in different conditions, its treatment remains similar.

However, this review has certain limitations. Only English-language studies were included. Further work can be done in the future that assesses more databases on top of PubMed and Embase and included non-English papers in a review.

One trait not included in this paper that has proven clinical evidence in chronic respiratory diseases would be vaccination status, including pneumococcal, influenza, and COVID-19 vaccination. Vaccination status has been widely shown to improve the clinical status of COPD patients, and it could potentially be an important trait classified under microbiological traits [62]. However, further studies are probably unnecessary, given the great importance of vaccination in overall health, beyond respiratory health in individuals [63].

## 5. Conclusions

The treatable traits paradigm has been shown to be a promising model in our efforts to improve care in chronic airway diseases, which are often complex and heterogenous in nature. With therapy tailored to the presence of treatable traits in each patient, greater personalisation of healthcare may be possible. Current literature suggests that a multidisciplinary approach with targeted therapies has the potential to be efficacious compared to the best practices currently implemented. Further studies exploring this novel approach will help assess its effectiveness in the real world, with the potential to significantly improve management of chronic airway diseases.

## Figures and Tables

**Figure 1 cells-10-03263-f001:**
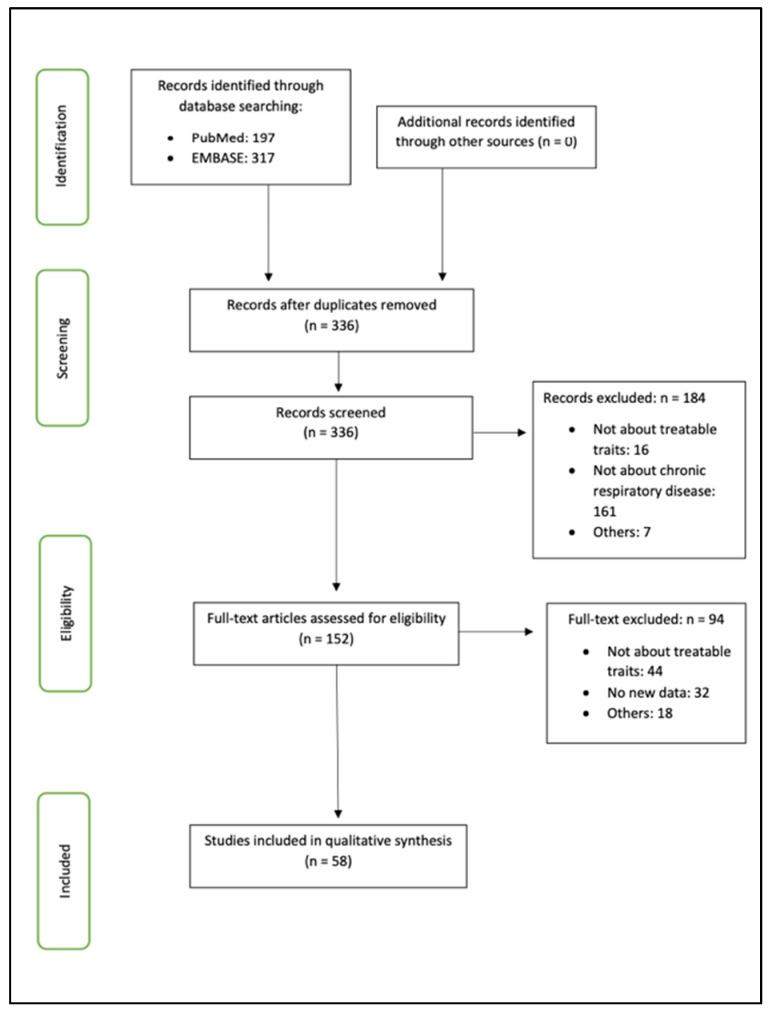
PRISMA flow diagram.

**Table 1 cells-10-03263-t001:** Quality assessment using the Newcastle-Ottawa quality assessment scale (NOS).

Study Sources	Representativeness	Selection of Non-Exposed Cohort	Ascertainment of Exposure	Outcome of Interest Not Present at Start of Study	Comparability	Assessment of Outcome	Adequacy of Follow-Up
Cazzola (2020) [3]		★	★	★		★	
Chalmers (2018) [4]		★	★	★	★★	★	
Connolly (2018) [5]		★		★	★		
Cottee (2020) [6]		★	★	★	★★	★	
Dean (2017) [7]		★		★	★		
Denton (2019) [8]		★	★	★	★★	★	
Descazeaux (2020) [9]		★	★	★	★	★	
Feng (2019) [10]		★	★	★	★	★	
Fingleton (2018) [11]		★	★	★	★	★	
Freitas (2020) [12]		★	★	★	★	★	
Garudadri (2018) [13]		★	★	★		★	
Gonçalves (2018) [14]		★	★	★	★	★	
Heffler (2019) [15]		★	★	★	★		
Higham (2019) [16]			★	★		★	
Hiles (2019) [17]		★	★	★	★★	★	
Hiles (2020) [18]				★			
Hinks (2020) [19]		★	★	★	★	★	
Honkoop (2019) [20]	★	★	★	★	★	★	
Jabeen (2018) [21]		★	★	★	★	★	
Koblizek (2019) [22]	★	★	★	★	★	★	
Kolmert (2019) [23]	★	★	★	★	★★	★	
Kuo (2019) [24]		★	★	★	★	★	
Lee (2020) [25]		★	★	★	★★	★	
Liu (2019) [26]		★	★	★	★	★	
Martin (2020) [1]			★	★	★	★	
Mathioudakis (2020) [27]		★	★	★	★★	★	
Matsunaga (2020) [28]		★	★	★	★★	★	
Matthes (2018) [29]			★	★	★	★	
McDonald (2019) [30]	★	★	★	★	★★	★	
McDonald (2019) [31]		★	★	★	★★	★	
Milne (2020) [32]				★			
Milne (2020) [33]	★			★			
Mohammed (2018) [34]		★	★	★		★	
Müllerová (2018) [35]	★	★	★	★		★	
Müllerová (2018) [36]	★	★	★	★		★	
Mummy (2020) [37]		★	★	★		★	
Osadnik (2019) [38]			★	★		★	
Papaioannou (2018) [39]				★			
Pavord (2020) [40]	★	★	★	★	★	★	
Llano (2019) [41]		★	★	★	★	★	
Llano (2020) [42]				★			
Ramsahai (2018) [43]				★			
Rosenkranz (2020) [44]				★			
Santos (2018) [45]	★			★			
Shoemark (2019) [46]		★	★	★		★	
Shteinberg (2020) [47]				★			
Simpson (2017) [48]				★			
Simpson (2018) [49]	★	★	★	★		★	
Soriano (2018) [50]		★	★	★	★★	★	
Tay (2018) [51]				★			
Tiew (2020) [52]			★	★	★★	★	
Tiotiu (2018) [53]		★	★	★	★	★	
Toledo-Pons (2019) [54]		★	★	★	★	★	
Meer (2019) [55]		★	★	★	★	★	
Dijk (2020) [56]			★	★		★	
Yii (2018) [57]			★	★		★	
Yii (2019) [58]		★	★	★		★	
Chung (2019) [59]		★	★	★		★	

A maximum of one star for each item within the Selection and Outcome categories. A maximum of two stars can be given for Comparability.

**Table 2 cells-10-03263-t002:** Overview of physiological treatable traits.

Condition	Treatable Trait	Trait-Identification Marker	Average Prevalence	Treatment Description and Benefits	Prognostic Implications	Author (Year)
Asthma	Airway limitation	Post-bronchodilator FEV1/FVC < 0.7 FEV1 < 80% predicted	52.5% (45.5–54.5%)	LAMA: ↑ Lung function, exacerbations Bronchial Thermoplasty: ↑ control, ↑ QoL, ↓ exacerbations SABA Short-acting anticholinergics: ↓ risk of admission Magnesium: ↓ odds of admission	Patients with poor PEFR response to salbutamol: ↑ airway obstruction, symptom duration and healthcare utilisation ↑ exacerbation risk	Hiles (2020) [18] Cazzola (2020) [3] Connolly (2018) [5] Simpson (2018) [49] Papaioannou (2018) [39] Hinks (2020) [19] Martin (2020) [1] McDonald (2019) [30]
Asthma	Hypoxemia/hypercapnia	SpO2 < 90% at rest or during 6 min walk test	10.9%	Investigation and implementation of domiciliary oxygen therapy and nasal CPAP	-	Hiles (2020) [18]
Asthma	Lung hyperinflation	>10% reduction in in inspiratory capacity	-	Systemic Corticosteroids: ↓ of dynamic hyperinflation	Dynamic hyperinflation ↑ in placebo group	Meer (2019) [55]
Bronchiectasis	Airway limitation	Low nasal NO, electron microscopic abnormalities, abnormal ciliary beating pattern	-	Inhaled saline, airway clearance, ongoing trial of ENaC inhibition	-	Shteinberg (2020) [47]
Bronchiectasis	Ciliary dysfunction	Elevated sweat chloride, characteristic electrophysiological abnormalities, CFTR mutations on two alleles	-	CFTR modulators	-	Shteinberg (2020) [47]
Chronic airway disease	Airway limitation	FEV1/FVC < 0.7 and FEV1 < 80% predicted	-	LAMA LABA-ICS: Significant functional and symptomatic improvement, Pulmonary rehabilitation	-	Llano (2020) [42]
COPD	Airway limitation	Post-bronchodilator FER < 70% and FEV1 < 80% predicted	88.9%	LAMA, LABA-ICS, Pulmonary rehabilitation	-	Hiles (2020) [18] Llano (2020) [42]
COPD	Hypoxemia/hypercapnia	PO2/PCO2	38.9%	Oxygen, NIV	Marker of poor prognosis	Llano (2020) [42] Gonçalves (2018) [14] Hiles (2020) [18]
COPD	Lung hyperinflation	RV > 175% predicted or RV/TLC ≥ 0.58	-	Endobronchial valves, coils: ↑ in lung function, ↓ dyspnoea, ↑ QoL, ↑ exercise tolerance, ↓ residual volume lung volume reduction surgery Bronchoscopic thermal vapour ablation: ↑ lung function and QoL in upper lobe prominent emphysema	-	Dijk (2020) [56]

Abbreviations: BMI, body mass index; CFTR, cystic fibrosis transmembrane conductance regulator; CPAP, continuous positive airway pressure; COPD, chronic obstructive pulmonary disease; EnaC, epithelial sodium channel; FER, forced expiratory ratio; FEV1, forced expiratory volume in the first second; FVC, forced vital capacity; ICS, inhaled corticosteroids; LABA, long-acting beta 2-agonists; LAMA, long-acting muscarinic antagonists; NIV, non-invasive ventilation; NO, nitric oxide; PCO2, partial pressure of carbon dioxide; PEFR, peak expiratory flow rate; QoL, quality of life; PO2, partial pressure of oxygen; RV, residual volume; SABA, short-acting beta-agonists; SpO2, peripheral capillary oxygen saturation; TLC, total lung capacity; ↑, increased/improved; ↓, decreased/reduced.

**Table 3 cells-10-03263-t003:** Overview of biochemical treatable traits.

Condition	Treatable Trait	Trait-Identification Marker	Prevalence (Range)	Treatment Description	Prognostic Implications	Author (Year)
Asthma	Eosinophilia	Blood/Sputum Eosinophilia	54.3% (51.4–56.4%)	Corticosteroids: ↑ FEV1 Omalizumab: Significant ↓ in exacerbations and ↑ in CARAT and AQLQ. FEV1 ↑, RV ↓, mean BE ↓ Mepolizumab/Anti IL-5 Anti IL4/IL -13: High FeNO responds to anti-IL4/IL13 therapies	Associated with severe asthma, frequent exacerbations, ↓ lung function at baseline	Chung (2019) [59] Connolly (2018) [5] Dean (2017) [7] Feng (2019) [10] Santos (2018) [45] Pavord (2020) [40] Llano (2019) [41] Papaioannou (2018) [39] Hiles (2020) [18] Hinks (2020) [19]
Asthma	FeNO	FeNO levels	-	FeNO-guided ICS treatment: Improved symptoms, ↑ asthma control, ↓ exacerbations, ↑ QoL		Dean (2017) [7] Honkoop (2019) [20] Kuo (2019) [24]
Asthma	Neutrophil elastase/inflammation; CXCR2R2	Sputum neutrophilis ≥ 61%	36.5% (27.3–40%)	Macrolides: ↓ exacerbation. May result in antibiotic resistance Smoking cessation: ↓ of neutrophilic inflammation, lung-function improvement in asthmatics	↑ exacerbation risk	Connolly (2018) [5] Dean (2017) [7] Simpson (2018) [49] Papaioannou (2018) [39] Hiles (2020) [18] Hinks (2020) [19] McDonald (2019) [30]
Asthma	Paucigranulocytic phenotype	Neutrophil levels <61% and eosinophil levels <2%	-	Macrolides, bronchodilators, bronchial thermoplasty	↑ risk of moderate-severe acute exacerbations, ↑ all-cause mortality. Higher airflow limitation and dyspnoea present in these patients.	Papaioannou (2018) [39]
Asthma	Proteins (periostin, galectin-3)	Sputum galectin-3	-	Anti-IgE therapy (omalizumab)	-	Dean (2017) [7]
Asthma	Type 2 inflammation	T2-high expression	42.0%	Salbutamol: Improved bronchodilator response Anti-T2 biologics: Major ↓ in severe exacerbations, small improvement in FEV1, improvement in asthma QoL scores	Corticosteroid insensitivity and oral corticosteroid dependence in severe patients	Chung (2019) [59] Simpson (2018) [49]
Bronchiectasis	Eosinophilia	IL-5, IL-13 and Gro-α in sputum	-	ICS, Bronchodilators, macrolides: Treatment showed little difference in clinical parameters between groups	-	Shteinburg (2020) [47] Shoemark (2019) [46]
Bronchiectasis	Neutrophil elastase/inflammation; CXCR2R2	Sputum neutrophils	-	Neutrophil elastase inhibitor: Significant ↑ in FEV1 and QoL. CXCR2 antagonist: ↓ sputum neutrophils, no difference in exacerbations. Macrolides Corticosteroids	↑ frequency of exacerbations and more rapid decline in FEV1 in some patients. Disease severity worse with ↑ BSI score, sputum volume, and ↓ predicted FEV1%	Chalmers (2018) [4] Shoemark (2019) [46]
Chronic airway disease	FeNO	Exhaled CO	-	Primary prevention	↑ acute exacerbations + major public health problem.	McDonald (2019) [31]
Chronic airway disease	T2-low inflammation	Eosinophil <100	-	Azithromycin, roflumilast, LABA-LAMA	-	Llano (2020) [42]
Chronic airway disease	Type 2 inflammation	Sputum/blood eosinophilia	-	ICS-LABA, ICS-LABA-LAMA, biologics	-	Llano (2020) [42] McDonald (2019) [31]
COPD	Eosinophilia	Sputum/blood eosinophilia	60.1% (22.2–60.1%)	Corticosteroids: Beneficial during exacerbations for patients with eosinophilia Anti-IL5	↑ number of moderate exacerbations, risk of future exacerbations. Exacerbations characterised by enhanced airway eosinophilic inflammation; generally milder, with ↓ mortality and ↓ hospital stay.	Garudadri (2018) [13] Gonçalves (2018) [14] Soriano (2018) [50] Müllerová (2018) [36] Müllerová (2018) [35] Hiles (2020) [18] Mathioudakis (2020) [27] Matsunaga (2020) [28] Matthes (2018) [29]
COPD	Neutrophil elastase/inflammation; CXCR2R2	Sputum neutrophils > 61%	44.4%	Macrolides	-	Hiles (2020) [18]
COPD	Proteins (periostin, galectin-3)	Specific marker	-	Specific therapy	-	Llano (2020) [42]
COPD	T2-low inflammation	Eosinophil < 100	-	Azithromycin, Roflumilast, LABA-LAMA	-	Llano (2020) [42]
COPD	Type 2 Inflammation	Eosinophil > 300/>100 if on OCS	-	ICS-LABA, ICS-LABA-LAMA, Biologics	-	Llano (2020) [42]
COPD	Vitamin D	Serum 25-hydroxycholecalciferol levels	-	Vitamin D supplementation: ↓ risk of respiratory tract infection.	VDD was associated with ↓ FEV1 at baseline and faster decline in FEV1	Llano (2020) [42]
Rhinitis/rhinosinusitis	Airway/nasal inflammation	Nasal cytology; nasal polyps biopsy	-	Corticosteroids, biologicals	-	Heffler (2019) [15]
United Airways Dz	Eosinophilia	Blood/sputum eosinophilia, blood periostin, high FeNO, absent specific IgE, non- reactive skin prick tests	-	Corticosteroids, anti-IL-5, IL-4, IL-13, anti-TSLP, CRTh2 antagonist	-	Yii (2018) [57]
United Airways Dz	Environmental exposure	Total IgE, skin prick tests Peak flow monitoring Specific bronchoprovocation challenge	-	Exposure avoidance, respiratory protection devices, anti-IgE	-	Yii (2018) [57]
United airways Dz	Neutrophil elastase/inflammation; CXCR2R2	IL-8, sputum neutrophilia	-	Smoking cessation, macrolides	-	Yii (2018) [57]

Abbreviations: AQLQ, Asthma Quality of Life Questionnaire; BE, base excess; BSI, bronchiectasis severity index; CARAT, Control of Allergic Rhinitis and Asthma Test; CO, carbon monoxide; CRTh2, prostaglandin D2 receptor 2; FeNO, fractional exhaled nitric oxide; FEV1, forced expiratory volume in the first second; FVC, forced vital capacity; ICS, inhaled corticosteroids; LABA, long-acting beta 2-agonists; LAMA, long-acting muscarinic antagonists; OCS, oral corticosteroids; QoL, quality of life; ↑, increase/improved; ↓, decreased/reduced.

**Table 4 cells-10-03263-t004:** Overview of psychosocial treatable traits.

Condition	Treatable Trait	Trait-Identification Marker	Prevalence (Range)	Treatment Description	Prognostic Implications	Author (Year)
Asthma	Adherence and technique	Adherence check Adherence rating scales	44.0% (26.9–61.8%)	Self-management education and WAP Treatment changed when possible to minimise devices Inhaler technique skills Self-management education with adherence-aiding strategies	Inhaler-device polypharmacy is one of the best predictors of exacerbation risk ↑ exacerbation risk	Connolly (2018) [5] Simpson (2018) [49] Hiles (2020) [18] McDonald (2019) [30]
Asthma	Smoking/ex-smoker	Medical history of smoking or exhaled CO ≥ 10 ppm	14.3% (13.9–14.5%)	Counseling and NRT or varenicline, bupropion	-	Connolly (2018) [5] Simpson (2018) [49] Hiles (2020) [18] Milne (2020) [32]
Chronic airway disease	Adherence and technique	Adherence check Adherence rating scales	-	Understanding reason for non-adherence, directing education of adherence-aiding strategies accordingly: Good adherence associated with ↓ severe exacerbations of asthma and COPD	Suboptimal inhaler technique and inhaler device polypharmacy associated with ↑ healthcare utilisation	Llano (2020) [42] McDonald (2019) [31]
Chronic airway disease	Smoking/ex-smoker	Medical history of smoking or exhaled CO ≥ 10 ppm	-	Counseling and NRT or varenicline, bupropion: Cessation ↓ lung function decline and future risk of exacerbations	Smoking is a risk factor for exacerbation	Llano (2020) [42] McDonald (2019) [31]
Chronic Airway Disease	Social issues	Interview	-	Activate support services	Poor family and social support and deprived socioeconomic status associated with ↑ symptom deterioration and exacerbation	McDonald (2019) [31]
COPD	Adherence and technique	Does not possess a WAP or does not use WAP during exacerbations Test of adherence to inhalers	55.6%	Self-management education and WAP Treatment changed when possible to minimise devices Inhaler technique skills Self-management education with adherence-aiding strategies	-	Hiles (2020) [18] Llano (2020) [42]
COPD	Smoking/ex-smoker	Medical history of smoking or exhaled CO ≥ 10 ppm	19.4%	Counseling and NRT or varenicline, bupropion	-	Hiles (2020) [18] Llano (2020) [42]

Abbreviations: CO, carbon monoxide; COPD, chronic obstructive pulmonary disease; NRT, nicotine replacement therapy; WAP, written action plan; ↑, increased/improved; ↓, decreased/reduced.

**Table 5 cells-10-03263-t005:** Overview of microbiological treatable traits.

Condition	Treatable Trait	Trait-Identification Marker	Prevalence (Range)	Treatment Description	Prognostic Implications	Author (Year)
Asthma	Chronic respiratory infection	History of chronic bronchitis, sputum analysis, culture/PCR	45.0% (34.8–47.3%)	Antibiotics: No treatment effect observed, asthma exacerbation Mucolytics Roflumilast Education Inhaled IFN-beta treatment: No effect on primary outcome, but morning PEFR increased in treatment group	H. influenzae is a clinically relevant pathogen in severe asthma that can be identified reliably using molecular microbiological methods	Connolly (2018) [5] Simpson (2018) [49] Hiles (2020) [18] Jabeen (2018) [21] Martin (2020) [1]
Asthma	Fungal colonisation	Total serum IgE measurement, serum Aspergillus IgE, Aspergillus skin testing	1.1%	Corticosteroids, anti-fungals, omalizumab	↑ exacerbation risk	Dean (2017) [7] Llano (2019) [41] McDonald (2019) [30]
Asthma	Microbial colonisation	Sputum culture +/− CT to exclude bronchiectasis	18.9% (12.7–55.6%)	Education, antibiotics (e.g., azithromycin), antibiotic-based WAP	-	Connolly (2018) [5] Hiles (2020) [18] Hinks (2020) [19]
Bronchiectasis	Chronic respiratory infection	Growth of pathogens in respiratory secretions	-	Antimicrobials	-	Shteinberg (2020) [47]
COPD	Chronic respiratory infection	Culture/PCR	55.6%	Antibiotics (Macrolides): Azithromycin: ↑ incidence of colonisation with macrolide-resistant organisms, an excessive rate of hearing decrements, and the prolongation of the QTc interval. ↓ ED visits and hospitalisation due to COPD exacerbations Education	Elevated SNOT-22 scores Produce colored/purulent sputum, even in stable state, have more severe dyspnea and impaired quality of life	Llano (2020) [42] Garudadri (2018) [13] Hiles (2020) [18] Koblizek (2019) [22] Matsunaga (2020) [28]
COPD	Microbial colonisation	Sputum culture Raised CRP Raised procalcitonin	44.8% (38.9–45%)	Antiviral vaccine and treatment (neuraminidase inhibitor) Education and antibiotics-based written plans Antibiotics: ↓ proportion of patients with exacerbations	↑ exacerbations, delayed recovery, and ↑ symptom burden ↓ mortality rate in exacerbations associated with respiratory viruses compared with those testing positive for bacteria either based on culture or PCR of sputum, endotracheal aspirates or BAL	Garudadri (2018) [13] Hiles (2020) [18] Mathioudakis (2020) [27]
United airways Dz	Fungal colonisation	Aspergillus skin test positivity, Aspergillus specific IgE, IgE > 1000 IU/mL, Blood eosinophil > 500/lL, precipitating antibodies to Aspergillus	-	Steroids, antifungal, anti-IgE	-	Yii (2018) [57]
United airways Dz	Staphylococcus aureus enterotoxin	Staph aureus enterotoxin IgE	-	Anti-IgE	-	Yii (2018) [57]
United airways Dz	URTI	Respiratory cultures and nucleic acid tests	-	Antivirals, antibiotics, vaccinations	-	Yii (2018) [57]

Abbreviations: BAL, bronchoalveolar lavage; COPD, chronic obstructive pulmonary disease; CRP, c-reactive protein; CT, computed tomography scan; ED, emergency department; IFN-beta, interferon-beta; PCR, polymerase chain reaction; PEFR, peak expiratory flow rate; SNOT-22, sinonasal outcome Test; WAP, written action plan; UnitedAirwayDz, united airway diseases; URTI, upper respiratory tract infection; ↑, increased/improved; ↓, decreased/reduced.

## Data Availability

Not applicable.

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
