# Peer review of "Treatable Traits in Chronic Respiratory Disease: A Comprehensive Review"

_cells, 2021, doi:10.3390/cells10113263_

Round 1

Reviewer 1 Report

Major comments:

The treatment guidelines of chronic respiratory diseases have relyed on severity and exacerbation-based approaches, so far. The authors propose a new personalized approach, termed the “treatable traits” approach to evaluate and establish targets of therapy in chronic respiratory diseases including asthma, COPD and bronchiectasis. For this purpose, they performed systematically review the current evidence regarding treatable traits in chronic respiratory diseases with the PubMed and EMBASE databases. Finally, they identified six themes as trait groups: physiological, biochemical, behavioural, microbiological, environmental, and comorbidity traits.

The author’s challenge of this manuscript compared to other papers regarding the treatable traits in asthma or COPD is the established 6 broad themes and the larger target area of disease. I would like to discuss these 2 issues as follows:

  1. Previous papers on treatable traits usually made 3 themes to classify treatable traits, which is ‘pulmonary trait’, ‘extrapulmonary trait’, and ‘behavioral traits/risk factors’. On the other hand, the authors newly established 6 broad themes containing ‘physiological trait’, ‘biochemical trait’, ‘behavioural trait’, ‘microbiological trait’, ‘environmental trait’, and ‘comorbidity trait’. I think the ordinary 3 themes is practical and useful. Especially, it is good for thinking about therapy. Conversely, the 6 themes is a bit too detailed but convenient for classifying experiment tools. This could prevent traits from being overlooked. But, for example, Nutrition/Weight, as well as Airway limitation and Lung hyperinflation, are categorized into ‘physiological trait’, that makes me feel a little uncomfortable. Also, ‘environmental trait’ has only 2 items. The authors should explain the reason for adopting this classification style, and describe the advantages of this classification over conventional ones.
  2. Previous papers on treatable traits usually studied in one defined disease area such as asthma, COPD or ACO, because they in previous papers were planning to study and prove the effectiveness of the methods. In the current paper, they make more broadly the target disease as chronic respiratory diseases including asthma, COPD and bronchiectasis. However, the number of traits for bronchiectasis is not so many. It is OK. The purpose of their setting this larger area could be to use it for others pulmonary disease as one of principle in the future. The authors should describe the reason of this setting.

              The treatable traits approach is a method like differentiation in mathematics, and the number of trait items could be unnecessarily large in this approach. For example, systemic or airway neutrophilic inflammation could often occur due to current smoking. In this case, quitting smoking is the best way, and macrolide or statin is not necessary. Furthermore, systemic exposure of polypharmacy is an important risk for adverse events and should be avoided. Therefore, after scoring each trait items, we will need to evaluate them thoroughly and integrate the unwanted ones. How do the authors think about that. The author also should discuss the pitfalls of thee treatable traits approach.

Minor comments:

  1. Is the information of the reference 9 correct? I can not reach the article with this title and publish date from PubMed or Home of Respirology. Maybe, that could be “McDonald VM, et al. Treatable traits can be identified in a severe asthma registry and predict future exacerbations. Respirology. 2019 Jan; 24(1): 37-47. doi: 10.1111/resp.13389. Epub 2018 Sep 19.”. Is that so?
  2. Maintenance of physical activity and vaccination against influenza, pneumococcus or SARS-CoV-2 are important in COPD treatment and are highly associated with prognosis of life. Physical activity should belong to ‘behavioural trait’, and vaccination should belong to ‘microbiological trait’ or ‘environmental trait’. These are not included in the papers reviewed by authors. However, they need to discuss them in the text.

Reviewer 2 Report

The authors performed a systematic review the current evidence regarding treatable traits in chronic respiratory diseases and to identify gaps in the current literature. Their findings suggest the potential of the multidisciplinary management with therapies targeted at treatable traits when added to the current best practices.

This systematic review paper is comprehensive and informative. The methods are well described and comply with reporting standards. There are clear pre-specified, eligibility criteria for studies being chosen or rejected for the review. The authors have made good effort to undertake thorough electronic searches using different databases. The results are clear and well presented. The conclusions appear to be supported by the study results. However, I have some comments that should be addressed by the authors before acceptance of the manuscript.

Fifty-eight studies were included in the systematic review. It is important that each individual study be cited in the manuscript. However, the bibliography consists of only 40 papers, many of which are review articles.

What does “review articles without new data” mean?

A systematic review is an analysis of the primary literature and, therefore, should include only original research articles (individual studies). Because review articles are considered as secondary sources, they should not be used in the data extraction process for the systematic review. However, review articles can be used to identify additional original research articles. Therefore, in the tables, results of individual studies only should be presented.

Round 2

Reviewer 2 Report

The authors addressed all my concerns. I have no further comments and recommend acceptance of the manuscripts in present form.